# On the Relationship Between Binary Classification, Bipartite Ranking, and Binary Class Probability Estimation

**Harikrishna Narasimhan     Shivani Agarwal**
Department of Computer Science and Automation
Indian Institute of Science, Bangalore 560012, India
`{harikrishna,shivani}@csa.iisc.ernet.in`

## Abstract

We investigate the relationship between three fundamental problems in machine learning: binary classification, bipartite ranking, and binary class probability estimation (CPE). It is known that a good binary CPE model can be used to obtain a good binary classification model (by thresholding at 0.5), and also to obtain a good bipartite ranking model (by using the CPE model directly as a ranking model); it is also known that a binary classification model does not necessarily yield a CPE model. However, not much is known about other directions. Formally, these relationships involve regret transfer bounds. In this paper, we introduce the notion of *weak* regret transfer bounds, where the mapping needed to transform a model from one problem to another depends on the underlying probability distribution (and in practice, must be estimated from data). We then show that, in this weaker sense, a good bipartite ranking model can be used to construct a good classification model (by thresholding at a suitable point), and more surprisingly, also to construct a good binary CPE model (by calibrating the scores of the ranking model).

## 1   Introduction

Learning problems with binary labels, where one is given training examples consisting of objects with binary labels (such as emails labeled spam/non-spam or documents labeled relevant/irrelevant), are widespread in machine learning. These include for example the three fundamental problems of *binary classification*, where the goal is to learn a classification model which, when given a new object, can predict its label; *bipartite ranking*, where the goal is to learn a ranking model that can rank new objects such that those in one category are ranked higher than those in the other; and *binary class probability estimation* (CPE), where the goal is to learn a CPE model which, when given a new object, can estimate the probability of its belonging to each of the two classes. Of these, binary classification is classical, although several fundamental questions related to binary classification have been understood only relatively recently [1–4]; bipartite ranking is more recent and has received much attention in recent years [5–8], and binary CPE, while a classical problem, also continues to be actively investigated [9,10]. All three problems abound in applications, ranging from email classification to document retrieval and computer vision to medical diagnosis.

It is well known that a good binary CPE model can be used to obtain a good binary classification model (in a formal sense that we will detail below; specifically, in terms of regret transfer bounds) [4, 11]; more recently, it was shown that a good binary CPE model can also be used to obtain a good bipartite ranking model (again, in terms of regret transfer bounds, to be detailed below) [12]. It is also known that a binary classification model cannot necessarily be converted to a CPE model.[1] However, beyond this, not much is understood about the exact relationship between these problems.[2]

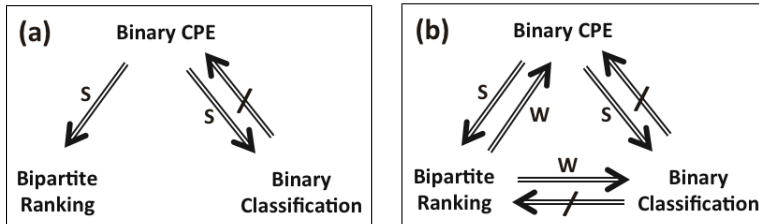

Figure 1: (a) Current state of knowledge; (b) State of knowledge after the results of this paper. Here 'S' denotes a 'strong' regret transfer relationship; 'W' denotes a 'weak' regret transfer relationship.

In this paper, we introduce the notion of *weak* regret transfer bounds, where the mapping needed to transform a model from one problem to another depends on the underlying probability distribution (and in practice, must be estimated from data). We then show such weak regret transfer bounds (under mild technical conditions) from bipartite ranking to binary classification, and from bipartite ranking to binary CPE. Specifically, we show that, given a good bipartite ranking model and access to either the distribution or a sample from it, one can estimate a suitable threshold and convert the ranking model into a good binary classification model; similarly, given a good bipartite ranking model and access to the distribution or a sample, one can 'calibrate' the ranking model to construct a good binary CPE model. Though weak, the regret bounds are non-trivial in the sense that the sample size required for constructing a good classification or CPE model from an existing ranking model is smaller than what might be required to learn such models from scratch.

The main idea in transforming a ranking model to a classifier is to find a threshold that minimizes the expected classification error on the distribution, or the empirical classification error on the sample. We derive these results for cost-sensitive classification with any cost parameter $c$. The main idea in transforming a ranking model to a CPE model is to find a monotonically increasing function from $\mathbb{R}$ to $[0, 1]$ which, when applied to the ranking model, minimizes the expected CPE error on the distribution, or the empirical CPE error on the sample; this is similar to the idea of isotonic regression [16–19]. The proof here makes use of a recent result of [20] which relates the squared error of a calibrated CPE model to classification errors over uniformly drawn costs, and a result on the Rademacher complexity of a class of bounded monotonically increasing functions on $\mathbb{R}$ [21]. As a by-product of our analysis, we also obtain a weak regret transfer bound from bipartite ranking to problems involving the area under the cost curve [22] as a performance measure.

The relationships between the three problems – both those previously known and those established in this paper – are summarized in Figure 1. As noted above, in a weak regret transfer relationship, given a model for one type of problem, one needs access to a data sample in order to transform this to a model for another problem. This is in contrast to the previous 'strong' relationships, where a binary CPE model can simply be thresholded at 0.5 (or cost $c$) to yield a classification model, or can simply be used directly as a ranking model. Nevertheless, even with the weak relationships, one still gets that a statistically consistent algorithm for bipartite ranking can be converted into a statistically consistent algorithm for binary classification or for binary CPE. Moreover, as we demonstrate in our experiments, if one has access to a good ranking model and only a small additional sample, then one is better off using this sample to transform the ranking model into a classification or CPE model rather than using the limited sample to learn a classification or CPE model from scratch.

The paper is structured as follows. We start with some preliminaries and background in Section 2. Sections 3 and 4 give our main results, namely weak regret transfer bounds from bipartite ranking to binary classification, and from bipartite ranking to binary CPE, respectively. Section 5 gives experimental results on both synthetic and real data. All proofs are included in the appendix.

## 2  Preliminaries and Background

Let $X$ be an instance space and let $D$ be a probability distribution on $X \times \{\pm 1\}$. For $(x, y) \sim D$, we denote $\eta(x) = \mathbf{P}(y = 1 \,|\, x)$ and $p = \mathbf{P}(y = 1)$. In the settings we are interested in, given a training sample $S = ((x_1, y_1), \ldots, (x_n, y_n)) \in (X \times \{\pm 1\})^n$ with examples drawn iid from $D$, the goal is to learn a binary classification model, a bipartite ranking model, or a binary CPE model. In what follows, for $u \in [-\infty, \infty]$, we will denote $\mathrm{sign}(u) = 1$ if $u > 0$ and $-1$ otherwise, and $\overline{\mathrm{sign}}(u) = 1$ if $u \geq 0$ and $-1$ otherwise.

**(Cost-Sensitive) Binary Classification.** Here the goal is to learn a model $h : X \to \{\pm 1\}$. Typically, one is interested in models $h$ with small expected 0-1 classification error:

$$\mathrm{er}_D^{0\text{-}1}[h] \;=\; \mathbf{E}_{(x,y)\sim D}\big[\mathbf{1}(h(x) \neq y)\big]\,,$$

where $\mathbf{1}(\cdot)$ is 1 if its argument is true and 0 otherwise; this is simply the probability that $h$ misclassifies an instance drawn randomly from $D$. The optimal 0-1 error (Bayes error) is

$$\mathrm{er}_D^{0\text{-}1,*} \;=\; \inf_{h:X\to\{\pm 1\}} \mathrm{er}_D^{0\text{-}1}[h] \;=\; \mathbf{E}_x\big[\min\big(\eta(x),\,1-\eta(x)\big)\big]\,;$$

this is achieved by the Bayes classifier $h^*(x) = \mathrm{sign}(\eta(x) - \frac{1}{2})$. The 0-1 classification regret of a classifier $h$ is then $\mathrm{regret}_D^{0\text{-}1}[h] \;=\; \mathrm{er}_D^{0\text{-}1}[h] - \mathrm{er}_D^{0\text{-}1,*}$. More generally, in a cost-sensitive binary classification problem with cost parameter $c \in (0,1)$, where the cost of a false positive is $c$ and that of a false negative is $(1-c)$, one is interested in models $h$ with small cost-sensitive 0-1 error:

$$\mathrm{er}_D^{0\text{-}1,c}[h] \;=\; \mathbf{E}_{(x,y)\sim D}\big[(1-c)\mathbf{1}\big(y=1, h(x)=-1\big) + c\,\mathbf{1}\big(y=-1, h(x)=1\big)\big]\,.$$

Note that for $c = \frac{1}{2}$, we get $\mathrm{er}_D^{0\text{-}1,\frac{1}{2}}[h] \;=\; \frac{1}{2}\mathrm{er}_D^{0\text{-}1}[h]$. The optimal cost-sensitive 0-1 error for cost parameter $c$ can then be seen to be

$$\mathrm{er}_D^{0\text{-}1,c,*} \;=\; \inf_{h:X\to\{\pm 1\}} \mathrm{er}_D^{0\text{-}1,c}[h] \;=\; \mathbf{E}_x\big[\min\big((1-c)\eta(x),\, c(1-\eta(x))\big)\big]\,;$$

this is achieved by the classifier $h_c^*(x) = \mathrm{sign}(\eta(x) - c)$. The $c$-cost-sensitive regret of a classifier $h$ is then $\mathrm{regret}_D^{0\text{-}1,c}[h] = \mathrm{er}_D^{0\text{-}1,c}[h] - \mathrm{er}_D^{0\text{-}1,c,*}$.

**Bipartite Ranking.** Here one wants to learn a ranking model $f : X \to \mathbb{R}$ that assigns higher scores to positive instances than to negative ones. Specifically, the goal is to learn a ranking function $f$ with small bipartite ranking error:

$$\mathrm{er}_D^{\mathrm{rank}}[f] = \mathbf{E}\Big[\mathbf{1}\big((y-y')(f(x)-f(x')) < 0\big) \;+\; \tfrac{1}{2}\,\mathbf{1}\big(f(x)=f(x')\big) \;\big|\; y \neq y'\Big]\,,$$

where $(x,y),(x',y')$ are assumed to be drawn iid from $D$; this is the probability that a randomly drawn pair of instances with different labels is mis-ranked by $f$, with ties broken uniformly at random. It is known that the ranking error of $f$ is equivalent to one minus the area under the ROC curve (AUC) of $f$ [5–7]. The optimal ranking error can be seen to be

$$\mathrm{er}_D^{\mathrm{rank},*} \;=\; \inf_{f:X\to\mathbb{R}} \mathrm{er}_D^{\mathrm{rank}}[f] = \frac{1}{2p(1-p)}\mathbf{E}_{x,x'}\Big[\min\big(\eta(x)(1-\eta(x')),\,\eta(x')(1-\eta(x))\big)\Big]\,;$$

this is achieved by any function $f^*$ that is a strictly monotonically increasing transformation of $\eta$. The ranking regret of a ranking function $f$ is given by $\mathrm{regret}_D^{\mathrm{rank}}[f] \;=\; \mathrm{er}_D^{\mathrm{rank}}[f] - \mathrm{er}_D^{\mathrm{rank},*}$.

**Binary Class Probability Estimation (CPE).** The goal here is to learn a class probability estimator or CPE model $\widehat{\eta} : X \to [0,1]$ with small squared error (relative to labels converted to $\{0,1\}$):

$$\mathrm{er}_D^{\mathrm{sq}}[\widehat{\eta}] \;=\; \mathbf{E}_{(x,y)\sim D}\Big[\big(\widehat{\eta}(x) - \tfrac{y+1}{2}\big)^2\Big]\,.$$

The optimal squared error can be seen to be

$$\mathrm{er}_D^{\mathrm{sq},*} \;=\; \inf_{\widehat{\eta}:X\to[0,1]} \mathrm{er}_D^{\mathrm{sq}}[\widehat{\eta}] \;=\; \mathrm{er}_D^{\mathrm{sq}}[\eta] \;=\; \mathbf{E}_x\big[\eta(x)(1-\eta(x))\big]\,.$$

The squared-error regret of a CPE model $\widehat{\eta}$ can be seen to be

$$\mathrm{regret}_D^{\mathrm{sq}}[\widehat{\eta}] \;=\; \mathrm{er}_D^{\mathrm{sq}}[\widehat{\eta}] - \mathrm{er}_D^{\mathrm{sq},*} \;=\; \mathbf{E}_x\big[\big(\widehat{\eta}(x) - \eta(x)\big)^2\big]\,.$$

**Regret Transfer Bounds.** The following (strong) regret transfer results from binary CPE to binary classification and from binary CPE to bipartite ranking are known:

**Theorem 1** ( [4, 11]). *Let $\widehat{\eta} : X \to [0,1]$. Let $c \in (0,1)$. Then the classifier $h(x) = \mathrm{sign}\big(\widehat{\eta}(x) - c\big)$ obtained by thresholding $\widehat{\eta}$ at $c$ satisfies*

$$\mathrm{regret}_D^{0\text{-}1,c}\big[\mathrm{sign}\circ(\widehat{\eta}-c)\big] \;\leq\; \mathbf{E}_x\big[|\widehat{\eta}(x) - \eta(x)|\big] \;\leq\; \sqrt{\mathrm{regret}_D^{\mathrm{sq}}[\widehat{\eta}]}\,.$$

**Theorem 2** ( [12]). *Let $\widehat{\eta} : X \to [0,1]$. Then using $\widehat{\eta}$ as a ranking model yields*

$$\mathrm{regret}_D^{\mathrm{rank}}[\widehat{\eta}] \;\leq\; \frac{1}{p(1-p)}\mathbf{E}_x\big[|\widehat{\eta}(x) - \eta(x)|\big] \;\leq\; \frac{1}{p(1-p)}\sqrt{\mathrm{regret}_D^{\mathrm{sq}}[\widehat{\eta}]}\,.$$

Note that as a consequence of these results, one gets that any learning algorithm that is statistically consistent for binary CPE, i.e. whose squared-error regret converges in probability to zero as the training sample size $n \to \infty$, can easily be converted into an algorithm that is statistically consistent for binary classification (with any cost parameter $c$, by thresholding the CPE models learned by the algorithm at $c$), or into an algorithm that is statistically consistent for bipartite ranking (by using the learned CPE models directly for ranking).

# 3 Regret Transfer Bounds from Bipartite Ranking to Binary Classification

In this section, we derive weak regret transfer bounds from bipartite ranking to binary classification. We derive two bounds. The first holds in an idealized setting where one is given a ranking model $f$ as well as access to the distribution $D$ for finding a suitable threshold to construct the classifier. The second bound holds in a setting where one is given a ranking model $f$ and a data sample $S$ drawn iid from $D$ for finding a suitable threshold; this bound holds with high probability over the draw of $S$. Our results will require the following assumption on the distribution $D$ and ranking model $f$:

**Assumption A.** *Let $D$ be a probability distribution on $X \times \{\pm 1\}$ with marginal distribution $\mu$ on $X$. Let $f : X \to \mathbb{R}$ be a ranking model, and let $\mu_f$ denote the induced distribution of scores $f(x) \in \mathbb{R}$ when $x \sim \mu$. We say $(D, f)$ satisfies Assumption A if $\mu_f$ is either discrete, continuous, or mixed with at most finitely many point masses.*

We will find it convenient to define the following set of all increasing functions from $\mathbb{R}$ to $\{\pm 1\}$:

$$\mathcal{T}_{\text{inc}} = \left\{ \theta : \mathbb{R} \to \{\pm 1\} \ : \ \theta(u) = \text{sign}(u - t) \text{ or } \theta(u) = \overline{\text{sign}}(u - t) \text{ for some } t \in [-\infty, \infty] \right\}.$$

**Definition 3 (Optimal classification transform).** *For any ranking model $f : X \to \mathbb{R}$, cost parameter $c \in (0, 1)$, and probability distribution $D$ over $X \times \{\pm 1\}$ such that $(D, f)$ satisfies Assumption A, define an optimal classification transform $\text{Thresh}_{D,f,c}$ as any increasing function from $\mathbb{R}$ to $\{\pm 1\}$ such that the classifier $h(x) = \text{Thresh}_{D,f,c}(f(x))$ resulting from composing $f$ with $\text{Thresh}_{D,f,c}$ yields minimum cost-sensitive 0-1 error on $D$:*

$$\text{Thresh}_{D,f,c} \in \text{argmin}_{\theta \in \mathcal{T}_{\text{inc}}} \left\{ \text{er}_D^{0\text{-}1,c} [\theta \circ f] \right\}. \tag{OP1}$$

We note that when $f$ is the class probability function $\eta$, we have $\text{Thresh}_{D,\eta,c}(u) = \text{sign}(u - c)$.

**Theorem 4 (Idealized weak regret transfer bound from bipartite ranking to binary classification based on distribution).** *Let $(D, f)$ satisfy Assumption A. Let $c \in (0, 1)$. Then the classifier $h(x) = \text{Thresh}_{D,f,c}(f(x))$ satisfies*

$$\text{regret}_D^{0\text{-}1,c} [\text{Thresh}_{D,f,c} \circ f] \leq \sqrt{2p(1-p) \, \text{regret}_D^{\text{rank}}[f]}.$$

In practice, one does not have access to the distribution $D$, and the optimal threshold must be estimated from a data sample. To this end, we define the following:

**Definition 5 (Optimal sample-based threshold).** *For any ranking model $f : X \to \mathbb{R}$, cost parameter $c \in (0, 1)$, and sample $S \in \cup_{n=1}^{\infty}(X \times \{\pm 1\})^n$, define an optimal sample-based threshold $\widehat{t}_{S,f,c}$ as any threshold on $f$ such that the resulting classifier $h(x) = \text{sign}(f(x) - \widehat{t}_{S,f,c})$ yields minimum cost-sensitive 0-1 error on $S$:*

$$\widehat{t}_{S,f,c} \in \text{argmin}_{t \in \mathbb{R}} \left\{ \text{er}_S^{0\text{-}1,c} [\text{sign} \circ (f - t)] \right\}, \tag{OP2}$$

*where $\text{er}_S^{0\text{-}1,c}[h]$ denotes the c-cost-sensitive 0-1 error of a classifier $h$ on the empirical distribution associated with $S$ (i.e. the uniform distribution over examples in $S$).*

Note that given a ranking function $f$, cost parameter $c$, and a sample $S$ of size $n$, the optimal sample-based threshold $\widehat{t}_{S,f,c}$ can be computed in $O(n \ln n)$ time by sorting the examples $(x_i, y_i)$ in $S$ based on the scores $f(x_i)$ and evaluating at most $n + 1$ distinct thresholds lying between adjacent score values (and above/below all score values) in this sorted order.

**Theorem 6 (Sample-based weak regret transfer bound from bipartite ranking to binary classification).** *Let $D$ be any probability distribution on $X \times \{\pm 1\}$ and $f : X \to \mathbb{R}$ be any fixed ranking model such that $(D, f)$ satisfies Assumption A. Let $S \in (X \times \{\pm 1\})^n$ be drawn randomly according to $D^n$. Let $c \in (0, 1)$. Let $0 < \delta \leq 1$. Then with probability at least $1 - \delta$ (over the draw of $S \sim D^n$), the classifier $h(x) = \text{sign}(f(x) - \widehat{t}_{S,f,c})$ obtained by thresholding $f$ at $\widehat{t}_{S,f,c}$ satisfies*

$$\text{regret}_D^{0\text{-}1,c} [\text{sign} \circ (f - \widehat{t}_{S,f,c})] \leq \sqrt{2p(1-p) \, \text{regret}_D^{\text{rank}}[f]} + \sqrt{\frac{32 \left( 2 \left( \ln(2n) + 1 \right) + \ln \left( \frac{4}{\delta} \right) \right)}{n}}.$$

The proof of Theorem 6 involves an application of the result in Theorem 4 together with a standard VC-dimension based uniform convergence result; specifically, the proof makes use of the fact that selecting the sample-based threshold in (OP2) is equivalent to empirical risk minimization over $\mathcal{T}_{\text{inc}}$. Note in particular that the above regret transfer bound, though 'weak', is non-trivial in that it suggests a good classifier can be constructed from a good ranking model using far fewer examples than might be required for learning a classifier from scratch based on standard VC-dimension bounds.

**Remark 7.** We note that, as a consequence of Theorem 6, one can use any learning algorithm that is statistically consistent for bipartite ranking to construct an algorithm that is consistent for (cost-sensitive) binary classification as follows: divide the training data into two (say equal) parts, use one part for learning a ranking model using the consistent ranking algorithm, and the other part for selecting a threshold on the learned ranking model; both terms in Theorem 6 will then go to zero as the training sample size increases, yielding consistency for (cost-sensitive) binary classification.

**Remark 8.** Another implication of the above result is a justification for the use of the AUC as a surrogate performance measure when learning in cost-sensitive classification settings where the misclassification costs are unknown during training time [23]. Here, instead of learning a classifier that minimizes the cost-sensitive classification error for a fixed cost parameter that may turn out to be incorrect, one can learn a ranking function with good ranking performance (in terms of AUC), and then later use a small additional sample to select a suitable threshold once the misclassification costs are known; the above result provides guarantees on the resulting classification performance in terms of the ranking (AUC) performance of the learned model.

## 4 Regret Transfer Bounds from Bipartite Ranking to Binary CPE

We now derive weak regret transfer bounds from bipartite ranking to binary CPE. Again, we derive two bounds: the first holds in an idealized setting where one is given a ranking model $f$ as well as access to the distribution $D$ for finding a suitable conversion to a CPE model; the second, which is a high-probability bound, holds in a setting where one is given a ranking model $f$ and a data sample $S$ drawn iid from $D$ for finding a suitable conversion. We will need the following definition:

**Definition 9** (**Calibrated CPE model**). *A binary CPE model $\widehat{\eta} : X \to [0, 1]$ is said to be* calibrated *w.r.t. a probability distribution $D$ on $X \times \{\pm 1\}$ if*

$$\mathbf{P}(y = 1 \mid \widehat{\eta}(x) = u) = u, \ \forall u \in \text{range}(\widehat{\eta}),$$

*where* $\text{range}(\widehat{\eta})$ *denotes the range of $\widehat{\eta}$.*

We will make use of the following result, which follows from results in [20] and shows that the squared error of a calibrated CPE model is related to the expected cost-sensitive error of a classifier constructed using the optimal threshold in Definition 3, over uniform costs in $(0, 1)$:

**Theorem 10** ( [20]). *Let $\widehat{\eta} : X \to [0, 1]$ be a binary CPE model that is calibrated w.r.t. $D$. Then*

$$\text{er}_D^{\text{sq}}[\widehat{\eta}] \ = \ 2 \, \mathbf{E}_{c \sim U(0,1)} \big[ \text{er}_D^{\text{0-1},c} \big[ \text{Thresh}_{D,\widehat{\eta},c} \circ \widehat{\eta} \big] \big] \,,$$

*where $U(0, 1)$ is the uniform distribution over $(0, 1)$ and $\text{Thresh}_{D,\widehat{\eta},c}$ is as defined in Definition 3.*

The proof of Theorem 10 follows from the fact that for any CPE model $\widehat{\eta}$ that is calibrated w.r.t. $D$, the optimal classification transform is given by $\text{Thresh}_{D,\widehat{\eta},c}(u) = \text{sign}(u - c)$, thus generalizing a similar result noted earlier for the true class probability function $\eta$.

We then have the following result, which shows that for a calibrated CPE model $\widehat{\eta} : X \to [0, 1]$, one can upper bound the squared-error regret in terms of the bipartite ranking regret; this result follows directly from Theorem 10 and Theorem 4:

**Lemma 11** (**Regret transfer bound for calibrated CPE models**). *Let $\widehat{\eta} : X \to [0, 1]$ be a binary CPE model that is calibrated w.r.t. $D$. Then*

$$\text{regret}_D^{\text{sq}}[\widehat{\eta}] \ \leq \ \sqrt{8p(1-p) \, \text{regret}_D^{\text{rank}}[\widehat{\eta}]} \,.$$

We are now ready to describe the construction of the optimal CPE transform in the idealized setting. We will find it convenient to define the following set:

$$\mathcal{G}_{\text{inc}} \ = \ \Big\{ g : \mathbb{R} \to [0, 1] \ : \ g \text{ is a monotonically increasing function} \Big\} \,.$$

**Definition 12** (**Optimal CPE transform**). *Let $f : X \to [a, b]$ (where $a, b \in \mathbb{R}$, $a < b$) be any bounded-range ranking model and $D$ be any probability distribution over $X \times \{\pm 1\}$ such that $(D, f)$ satisfies Assumption A. Moreover assume that $\mu_f$ (see Assumption A), if mixed, does not have a point mass at the end-points $a, b$, and that the function $\eta_f : [a, b] \to [0, 1]$ defined as $\eta_f(t) = \mathbf{P}(y = 1 \mid f(x) = t)$ is square-integrable w.r.t. the density of the continuous part of $\mu_f$. Define an* optimal CPE transform $\text{Cal}_{D,f}$ *as any monotonically increasing function from $\mathbb{R}$ to $[0, 1]$ such that the CPE model $\widehat{\eta}(x) = \text{Cal}_{D,f}(f(x))$ resulting from composing $f$ with $\text{Cal}_{D,f}$ yields minimum squared error on $D$ (see appendix for existence of $\text{Cal}_{D,f}$ under these conditions):*

$$\text{Cal}_{D,f} \in \text{argmin}_{g \in \mathcal{G}_{\text{inc}}} \big\{ \text{er}_D^{\text{sq}} \big[ g \circ f \big] \big\} \,. \tag{OP3}$$

**Lemma 13** (**Properties of $\mathrm{Cal}_{D,f}$**). *Let $(D, f)$ satisfy the conditions of Definition 12. Then*

1. *$(\mathrm{Cal}_{D,f} \circ f)$ is calibrated w.r.t. $D$.*

2. *$\mathrm{er}_D^{\mathrm{rank}}\big[\mathrm{Cal}_{D,f} \circ f\big] \leq \mathrm{er}_D^{\mathrm{rank}}[f].$*

The proof of Lemma 13 is based on equivalent results for the minimizer of a sample version of (OP3) [24, 25]. Combining this with Lemma 11 immediately gives the following result:

**Theorem 14** (**Idealized weak regret transfer bound from bipartite ranking to binary CPE based on distribution**). *Let $(D, f)$ satisfy the conditions of Definition 12. Then the CPE model $\widehat{\eta}(x) = \mathrm{Cal}_{D,f}(f(x))$ obtained by composing $f$ with $\mathrm{Cal}_{D,f}$ satisfies*

$$\mathrm{regret}_D^{\mathrm{sq}}\big[\mathrm{Cal}_{D,f} \circ f\big] \leq \sqrt{8p(1-p)\,\mathrm{regret}_D^{\mathrm{rank}}[f]}.$$

We now derive a sample version of the above result.

**Definition 15** (**Optimal sample-based CPE transform**). *For any ranking model $f : X \to \mathbb{R}$ and sample $S \in \cup_{n=1}^{\infty}(X \times \{\pm 1\})^n$, define an* optimal sample-based transform *$\widehat{\mathrm{Cal}}_{S,f}$ as any monotonically increasing function from $\mathbb{R}$ to $[0, 1]$ such that the CPE model $\widehat{\eta}(x) = \widehat{\mathrm{Cal}}_{S,f}(f(x))$ resulting from composing $f$ with $\widehat{\mathrm{Cal}}_{S,f}$ yields minimum squared error on $S$:*

$$\widehat{\mathrm{Cal}}_{S,f} \in \mathrm{argmin}_{g \in \mathcal{G}_{\mathrm{inc}}}\big\{\mathrm{er}_S^{\mathrm{sq}}\big[g \circ f\big]\big\}, \tag{OP4}$$

*where $\mathrm{er}_S^{\mathrm{sq}}[\widehat{\eta}]$ denotes the squared error of a CPE model $\widehat{\eta}$ on the empirical distribution associated with $S$ (i.e. the uniform distribution over examples in $S$).*

The above optimization problem corresponds to the well-known *isotonic regression* problem and can be solved in $O(n \ln n)$ time using the pool adjacent violators (PAV) algorithm [16] (the PAV algorithm outputs a score in $[0, 1]$ for each instance in $S$ such that these scores preserve the ordering of $f$; a straightforward interpolation of the scores then yields a monotonically increasing function of $f$). We then have the following sample-based weak regret transfer result:

**Theorem 16** (**Sample-based weak regret transfer bound from bipartite ranking to binary CPE**). *Let $D$ be any probability distribution on $X \times \{\pm 1\}$ and $f : X \to [a, b]$ be any fixed ranking model such that $(D, f)$ satisfies the conditions of Definition 12. Let $S \in (X \times \{\pm 1\})^n$ be drawn randomly according to $D^n$. Let $0 < \delta \leq 1$. Then with probability at least $1 - \delta$ (over the draw of $S \sim D^n$), the CPE model $\widehat{\eta}(x) = \widehat{\mathrm{Cal}}_{S,f}(f(x))$ obtained by composing $f$ with $\widehat{\mathrm{Cal}}_{S,f}$ satisfies*

$$\mathrm{regret}_D^{\mathrm{sq}}\big[\widehat{\mathrm{Cal}}_{S,f} \circ f\big] \leq \sqrt{8p(1-p)\,\mathrm{regret}_D^{\mathrm{rank}}[f]} + 96\sqrt{\frac{2\ln(n)}{n}} + 2\sqrt{\frac{2\ln\left(\frac{8}{\delta}\right)}{n}}.$$

The proof of Theorem 16 involves an application of the idealized result in Theorem 14, together with a standard uniform convergence argument based on Rademacher averages applied to the function class $\mathcal{G}_{\mathrm{inc}}$; for this, we make use of a result on the Rademacher complexity of this class [21].

**Remark 17.** As in the case of binary classification, we note that, as a consequence of Theorem 16, one can use any learning algorithm that is statistically consistent for bipartite ranking to construct an algorithm that is consistent for binary CPE as follows: divide the training data into two (say equal) parts, use one part for learning a ranking model using the consistent ranking algorithm, and the other part for selecting a CPE transform on the learned ranking model; both terms in Theorem 16 will then go to zero as the training sample size increases, yielding consistency for binary CPE.

**Remark 18.** We note a recent result in [19] giving a bound on the empirical squared error of a CPE model constructed from a ranking model using isotonic regression in terms of the empirical ranking error of the ranking model. However, this does not amount to a regret transfer bound.

**Remark 19.** Finally, we note that the quantity $\mathbf{E}_{c \sim U(0,1)}\big[\mathrm{er}_D^{0\text{-}1,c}\big[\mathrm{Thresh}_{D,\widehat{\eta},c} \circ \widehat{\eta}\big]\big]$ that appears in Theorem 10 is also the area under the cost curve [20, 22]; since this quantity is upper bounded in terms of $\mathrm{regret}_D^{\mathrm{rank}}[f]$ by virtue of Theorem 4, we also get a weak regret transfer bound from bipartite ranking to problems where the area under the cost curve is a performance measure of interest. In particular, this implies that algorithms that are statistically consistent with respect to AUC can also be used to construct algorithms that are statistically consistent w.r.t. the area under the cost curve.

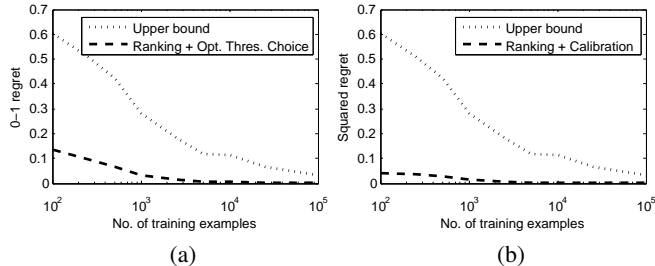

Figure 2: Results on synthetic data. A ranking model was learned using a pairwise linear logistic regression ranking algorithm (which is a consistent ranking algorithm for the distribution used in these experiments); this was followed by an optimal choice of classification threshold (with $c = \frac{1}{2}$) or optimal CPE transform based on the distribution as outlined in Sections 3 and 4. The plots show (a) 0-1 classification regret of the resulting classification model together with the corresponding upper bound from Theorem 4; and (b) squared-error regret of the resulting CPE model together with the corresponding upper bound from Theorem 14. As can be seen, in both cases, the classification/CPE regret converges to zero as the training sample size increases.

## 5 Experiments

We conducted two types of experiments to evaluate the results described in this paper: the first involved synthetic data drawn from a known distribution for which the classification and ranking regrets could be calculated exactly; the second involved real data from the UCI Machine Learning Repository. In the first experiment, we learned ranking models using a consistent ranking algorithm on increasing training sample sizes, converted the learned models using the optimal threshold or CPE transforms described in Sections 3 and 4 based on the distribution, and verified that this yielded classification and CPE models with 0-1 classification regret and squared-error regret converging to zero. In the second experiment, we simulated a setting where a ranking model has been learned from some data, the original training data is no longer available, and a classification/CPE model is needed; we investigated whether in such a setting the ranking model could be used in conjunction with a small additional data sample to produce a useful classification or CPE model.

### 5.1 Synthetic Data

Our first goal was to verify that using ranking models learned by a statistically consistent ranking algorithm and applying the distribution-based transformations described in Sections 3 and 4 yields classification/CPE models with classification/CPE regret converging to zero. For these experiments, we generated examples in $(X = \mathbb{R}^d) \times \{\pm 1\}$ (with $d = 100$) as follows: each example was assigned a positive/negative label with equal probability, with the positive instances drawn from a multivariate Gaussian distribution with mean $\mu \in \mathbb{R}^d$ and covariance matrix $\Sigma \in \mathbb{R}^{d \times d}$, and negative instances drawn from a multivariate Gaussian distribution with mean $-\mu$ and the same covariance matrix $\Sigma$; here $\mu$ was drawn uniformly at random from $\{-1, 1\}^d$, and $\Sigma$ was drawn from a Wishart distribution with 200 degrees of freedom and a randomly drawn invertible PSD scale matrix. For this distribution, the optimal ranking and classification models are linear. Training samples of various sizes $n$ were generated from this distribution; in each case, a linear ranking model was learned using a pairwise linear logistic regression algorithm (with regularization parameter set to $1/\sqrt{n}$), and an optimal threshold (with $c = \frac{1}{2}$) or CPE transform was then applied to construct a binary classification or CPE model. In this case the ranking regret and 0-1 classification regret of a linear model and can be computed exactly; the squared-error regret for the CPE model was computed approximately by sampling instances from the distribution. The results are shown in Figure 2. As can be seen, the classification and squared-error regrets of the classification and CPE models constructed both satisfy the bounds from Theorems 4 and 14, and converge to zero as the bounds suggest.

### 5.2 Real Data

Our second goal was to investigate whether good classification and CPE models can be constructed in practice by applying the data-based transformations described in Sections 3 and 4 to an existing ranking model. For this purpose, we conducted experiments on several data sets drawn from the UCI Machine Learning Repository[3]. We present representative results on two data sets: Spambase (4601

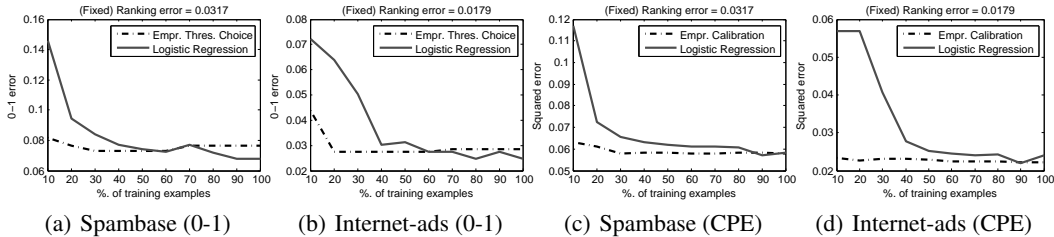

| (a) Spambase (0-1) | (b) Internet-ads (0-1) | (c) Spambase (CPE) | (d) Internet-ads (CPE) |

Figure 3: Results on real data from the UCI repository. A ranking model was learned using a pairwise linear logistic regression ranking algorithm from a part of the data set that was then discarded. The remaining data was divided into training and test sets. The training data was then used to estimate an empirical (sample-based) classification threshold and CPE transform (calibration) for this ranking model as outlined in Sections 3 and 4. Using the same training data, a binary classifier and CPE model were also learned from scratch using a standard linear logistic regression algorithm. The plots show the resulting test error for both approaches. As can be seen, if only a small amount of additional data is available, then using this data to convert an existing ranking model into a classification/CPE model is more beneficial than learning a classification/CPE model from scratch.

instances, 57 features) and Internet Ads (3279 instances, 1554 features[4]). Here we divided each data set into three equal parts. One part was used to learn a ranking model using a pairwise linear logistic regression algorithm, and was then discarded. This allowed us to simulate a situation where a (reasonably good) ranking model is available, but the original training data used to learn the model is no longer accessible. Various subsets of the second part of the data (of increasing size) were then used to estimate a data-based threshold or CPE transform on this ranking model using the optimal sample-based methods described in Sections 3 and 4. The performance of the constructed classification and CPE models on the third part of the data, which was held out for testing purposes, is shown in Figure 3. For comparison, we also show the performance of binary classification and CPE models learned directly from the same subsets of the second part of the data using a standard linear logistic regression algorithm. In each case, the regularization parameter for both standard logistic regression and pairwise logistic regression was chosen from $\{10^{-4}, 10^{-3}, 10^{-2}, 10^{-1}, 1, 10, 10^2\}$ using 5-fold cross validation on the corresponding training data. As can be seen, when one has access to a previously learned (or otherwise available) ranking model with good ranking performance, and only a small amount of additional data, then one is better off using this data to estimate a threshold/CPE transform and converting the ranking model into a classification/CPE model, than learning a classification/CPE model from this data from scratch. However, as can also be seen, the eventual performance of the classification/CPE model thus constructed is limited by the ranking performance of the original ranking model; therefore, once there is sufficient additional data available, it is advisable to use this data to learn a new model from scratch.

## 6 Conclusion

We have investigated the relationship between three fundamental problems in machine learning: binary classification, bipartite ranking, and binary class probability estimation (CPE). While formal regret transfer bounds from binary CPE to binary classification and to bipartite ranking are known, little has been known about other directions. We have introduced the notion of *weak* regret transfer bounds that require access to a distribution or data sample, and have established the existence of such bounds from bipartite ranking to binary classification and to binary CPE. The latter result makes use of ideas related to calibration and isotonic regression; while these ideas have been used to calibrate scores from real-valued classifiers to construct probability estimates in practice, to our knowledge, this is the first use of such ideas in deriving formal regret bounds in relation to ranking. Our experimental results demonstrate possible uses of the theory developed here.

**Acknowledgments**

Thanks to Karthik Sridharan for pointing us to a result on monotonically increasing functions. Thanks to the anonymous reviewers for many helpful suggestions. HN gratefully acknowledges support from a Google India PhD Fellowship. SA thanks the Department of Science & Technology (DST), the Indo-US Science & Technology Forum (IUSSTF), and Yahoo! for their support.

## Footnotes

[1]Note that we start from a *single* classification model, which rules out the probing reduction of [13].

[2]There are some results suggesting equivalence between specific boosting-style classification and ranking algorithms [14,15], but this does not say anything about relationships between the problems *per se*.

[3] http://archive.ics.uci.edu/ml/

[4]The original data set contains 1558 features; we discarded 4 features with missing entries.

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
