[Supplementary Material]

# On the Relationship Between Binary Classification, Bipartite Ranking, and Binary Class Probability Estimation
## Appendix

## A  Proof of Theorem 4

*Proof.* Assume w.l.o.g. that $\text{Thresh}_{D,f,c}(u) = \text{sign}(u - t^*)$ for some $t^* \in [-\infty, \infty]$; a similar analysis can be shown when $\text{Thresh}_{D,f,c}(u) = \overline{\text{sign}}(u - t^*)$ for some $t^*$. We first recall the following result of Clémençon et al. [8] (adapted as in [26] to account for ties and conditioning on $y \neq y'$).

$$\text{regret}_D^{\text{rank}}[f] \;\;=\;\; \frac{1}{2p(1-p)}\mathbf{E}_{x,x'}\Big[\big|\eta(x) - \eta(x')\big|\Big(\mathbf{1}\big((f(x) - f(x'))(\eta(x) - \eta(x')) < 0\big)$$
$$+ \frac{1}{2}\mathbf{1}\big(f(x) = f(x')\big)\Big)\Big].$$

Next, given a binary classifier $h : X \to \{\pm 1\}$ and a cost parameter $c \in (0,1)$, the cost-sensitive classification error can be rewritten as

$$\text{er}_D^{0\text{-}1,c}[h] = \mathbf{E}_x\big[(1-c)\eta(x)\mathbf{1}\big(h(x) = -1\big) + c\big(1 - \eta(x)\big)\mathbf{1}\big(h(x) = 1\big)\big]$$

and the corresponding regret can be expanded as

$$\begin{aligned}
&\text{regret}_D^{0\text{-}1,c}[h] \\
&=\;\; \mathbf{E}_x\big[(1-c)\eta(x)\mathbf{1}\big(h(x) = -1\big) + c\big(1 - \eta(x)\big)\mathbf{1}\big(h(x) = 1\big)\big] \\
&\qquad\qquad\qquad -\; \mathbf{E}_x\big[(1-c)\eta(x)\mathbf{1}\big(\eta(x) \leq c\big) + c\big(1 - \eta(x)\big)\mathbf{1}\big(\eta(x) > c\big)\big] \\
&=\;\; \mathbf{E}_x\big[(c - \eta(x))\mathbf{1}\big(h(x) = 1,\; \eta(x) \leq c\big)\big] + \mathbf{E}_x\big[(\eta(x) - c)\mathbf{1}\big(h(x) = -1,\; \eta(x) > c\big)\big].
\end{aligned}$$

For $h = \text{sign} \circ (f - t^*)$,

$$\begin{aligned}
&\text{regret}_D^{0\text{-}1,c}[\text{sign} \circ (f - t^*)] \\
&=\;\; \mathbf{E}_x\big[(c - \eta(x))\mathbf{1}\big(f(x) > t^*,\; \eta(x) \leq c\big)\big] + \mathbf{E}_x\big[(\eta(x) - c)\mathbf{1}\big(f(x) \leq t^*,\; \eta(x) > c\big)\big] \quad (1) \\
&=\;\; a + b \;\;\text{(say)}.
\end{aligned}$$

We then have

$$\begin{aligned}
2p(1-p)\,\text{regret}_D^{\text{rank}}[f] \;\geq\;\; & \frac{1}{2}\mathbf{E}_{x,x'}\Big[\big|\eta(x) - \eta(x')\big|\Big(\mathbf{1}\big((f(x) - f(x'))(\eta(x) - \eta(x')) \leq 0\big)\Big)\Big] \\
& \text{(getting rid of the term accounting for ties)} \\
\geq\;\; & \frac{1}{2}\mathbf{E}_{x,x'}\Big[\big|\eta(x) - \eta(x')\big|\Big(\mathbf{1}\big(f(x) \geq f(x'),\; \eta(x) \leq c,\; \eta(x') > c\big) \\
& \qquad\qquad\qquad\qquad + \mathbf{1}\big(f(x) \leq f(x'),\; \eta(x) > c,\; \eta(x') \leq c\big)\Big)\Big] \\
=\;\; & \frac{2}{2}\mathbf{E}_{x,x'}\Big[\big|\eta(x) - \eta(x')\big|\Big(\mathbf{1}\big(f(x) \geq f(x'),\; \eta(x) \leq c,\; \eta(x') > c\big)\Big)\Big] \\
=\;\; & \text{term}_1 + \text{term}_2 + \text{term}_3, \hspace{4cm} (2)
\end{aligned}$$

where

$$\begin{aligned}
\text{term}_1 \;&=\; \mathbf{E}_{x,x'}\Big[\big|\eta(x) - \eta(x')\big|\Big(\mathbf{1}\big(f(x) \geq f(x') > t^*,\; \eta(x) \leq c,\; \eta(x') > c\big)\Big)\Big], \\
\text{term}_2 \;&=\; \mathbf{E}_{x,x'}\Big[\big|\eta(x) - \eta(x')\big|\Big(\mathbf{1}\big(t^* \geq f(x) \geq f(x'),\; \eta(x) \leq c,\; \eta(x') > c\big)\Big)\Big] \;\;\text{and} \\
\text{term}_3 \;&=\; \mathbf{E}_{x,x'}\Big[\big|\eta(x) - \eta(x')\big|\Big(\mathbf{1}\big(f(x) > t^*,\; f(x') \leq t^*,\; \eta(x) \leq c,\; \eta(x') > c\big)\Big)\Big].
\end{aligned}$$

Each of the above terms corresponds to different sets of pairs of instances; $\text{term}_1$ corresponds to pairs where both instances are ranked by $f$ above $t^*$; $\text{term}_2$ corresponds to pairs where both instances are

ranked by $f$ below (or at the same position as) $t^*$; $\text{term}_3$ corresponds to pairs $(x, x')$, where $x$ is ranked by $f$ above $t^*$, while $x'$ is ranked below (or at the same position as) $t^*$. We next bound each of these terms separately.

$\text{term}_1$

$$
\begin{aligned}
&= \mathbf{E}_{x,x'}\Big[\big|\eta(x') - c + c - \eta(x)\big|\Big(\mathbf{1}\big(f(x) \geq f(x') > t^*,\ \eta(x) \leq c,\ \eta(x') > c\big)\Big)\Big] \\
&\geq \mathbf{E}_{x,x'}\Big[2\big|\eta(x') - c\big|\big|c - \eta(x)\big|\Big(\mathbf{1}\big(f(x) \geq f(x') > t^*,\ \eta(x) \leq c,\ \eta(x') > c\big)\Big)\Big] \\
&\qquad\qquad\qquad\qquad\qquad\qquad \text{(since } u + v \geq 2\sqrt{uv} \geq 2uv,\ \forall u, v \in [0,1]) \\
&= 2\mathbf{E}_x\Big[\big|c - \eta(x)\big|\mathbf{1}(f(x) > t^*,\ \eta(x) \leq c)\mathbf{E}_{x'}\big[\big|\eta(x') - c\big|\mathbf{1}\big(t^* < f(x') \leq f(x),\ \eta(x') > c\big)\big]\Big].
\end{aligned}
$$
(3)

By definition, $t^*$ yields the minimum classification regret among all choices of thresholds $t \in \mathbb{R}$:

$$
\begin{aligned}
t^* &= \underset{t \in [-\infty, \infty]}{\operatorname{argmin}} \Big\{\text{regret}_D^{\text{0-1},c}\big[\text{sign} \circ (f - t)\big]\Big\} \\
&= \underset{t \in [-\infty, \infty]}{\operatorname{argmin}}\ \mathbf{E}_{x'}\Big[\big(\eta(x') - c\big)\mathbf{1}\big(f(x') \leq t,\ \eta(x') > c\big) + \big(c - \eta(x')\big)\mathbf{1}\big(f(x') > t,\ \eta(x') \leq c\big)\Big]
\end{aligned}
$$
(from Eq. (1)).

It can hence be shown that for any $t > t^*$,
$\mathbf{E}_{x'}\big[\big|\eta(x') - c\big|\mathbf{1}\big(t^* < f(x') \leq t,\ \eta(x') > c\big)\big] \geq \mathbf{E}_{x'}\big[\big|c - \eta(x')\big|\mathbf{1}\big(t^* < f(x') \leq t,\ \eta(x') \leq c\big)\big]$.
Applying the above inequality to Eq. (3) with $t = f(x)$, we have

$\text{term}_1$

$$
\begin{aligned}
&\geq 2\mathbf{E}_x\Big[\big|c - \eta(x)\big|\mathbf{1}(f(x) > t^*,\ \eta(x) \leq c)\mathbf{E}_{x'}\big[\big|c - \eta(x')\big|\mathbf{1}\big(t^* < f(x') \leq f(x),\ \eta(x') \leq c\big)\big]\Big] \\
&\geq \frac{2}{2}\mathbf{E}_x\Big[\big|c - \eta(x)\big|\mathbf{1}(f(x) > t^*,\ \eta(x) \leq c)\mathbf{E}_{x'}\big[\big|c - \eta(x')\big|\mathbf{1}\big(t^* < f(x'),\ \eta(x') \leq c\big)\big]\Big] \\
&\qquad\qquad\qquad \Big(\text{since } \mathbf{E}_{x,x'}[g(x, x')\mathbf{1}(f(x) \leq f(x'))] \geq \frac{1}{2}\mathbf{E}_{x,x'}[g(x, x')]\Big) \\
&= \mathbf{E}_x\Big[\big|c - \eta(x)\big|\mathbf{1}(f(x) > t^*,\ \eta(x) \leq c)\Big]\mathbf{E}_{x'}\Big[\big|c - \eta(x')\big|\mathbf{1}\big(t^* < f(x'),\ \eta(x') \leq c\big)\Big] \\
&= \mathbf{E}_x\Big[\big|c - \eta(x)\big|\mathbf{1}(f(x) > t^*,\ \eta(x) \leq c)\Big]^2 \\
&= a^2.
\end{aligned}
$$

Similarly, one can show

$$
\text{term}_2 \geq \mathbf{E}_x\Big[\big|\eta(x) - c\big|\mathbf{1}(f(x) \leq t^*,\ \eta(x) > c)\Big]^2 = b^2.
$$

In the case of $\text{term}_3$, we have

$$
\begin{aligned}
\text{term}_3 &= \mathbf{E}_{x,x'}\Big[\big|\eta(x') - c + c - \eta(x)\big|\Big(\mathbf{1}\big(f(x) > t^*,\ f(x') \leq t^*,\ \eta(x) \leq c,\ \eta(x') > c\big)\Big)\Big] \\
&\geq \mathbf{E}_{x,x'}\Big[2\big|\eta(x') - c\big|\big|c - \eta(x)\big|\Big(\mathbf{1}\big(f(x) > t^*,\ f(x') \leq t^*,\ \eta(x) \leq c,\ \eta(x') > c\big)\Big)\Big] \\
&\qquad\qquad\qquad\qquad\qquad \text{(since } u + v \geq 2\sqrt{uv} \geq 2uv,\ \forall u, v \in [0,1]) \\
&\geq 2\mathbf{E}_{x,x'}\Big[\big|c - \eta(x)\big|\mathbf{1}\big(f(x) > t^*,\ \eta(x) \leq c\big)\big|\eta(x') - c\big|\mathbf{1}\big(f(x') \leq t^*,\ \eta(x') > c\big)\Big] \\
&= 2\mathbf{E}_x\Big[\big|c - \eta(x)\big|\mathbf{1}\big(f(x) > t^*,\ \eta(x) \leq c\big)\Big]\mathbf{E}_{x'}\Big[\big|\eta(x') - c\big|\mathbf{1}\big(f(x') \leq t^*,\ \eta(x') > c\big)\Big] \\
&= 2ab.
\end{aligned}
$$

Applying the bounds on $\text{term}_1$, $\text{term}_2$ and $\text{term}_3$ in Eq. (2), we have

$$
\begin{aligned}
2p(1 - p)\,\text{regret}_D^{\text{rank}}[f] &\geq a^2 + b^2 + 2ab \\
&= (a + b)^2 \\
&= \big(\text{regret}_D^{\text{0-1},c}[\text{sign} \circ (f - t^*)]\big)^2.
\end{aligned}
$$

Hence the proof. $\qquad\qquad\qquad\qquad\qquad\qquad\qquad\qquad\qquad\qquad\qquad\qquad\qquad\qquad$ $\square$

# B  Proof of Theorem 6

*Proof.*

$$\begin{aligned}
&\text{regret}_D^{0\text{-}1,c}[\text{sign} \circ (f - \widehat{t}_{S,f,c})] \\
&= \text{er}_D^{0\text{-}1,c}[\text{sign} \circ (f - \widehat{t}_{S,f,c})] - \text{er}_D^{0\text{-}1,c,*} \\
&= \text{er}_D^{0\text{-}1,c}[\text{sign} \circ (f - \widehat{t}_{S,f,c})] - \text{er}_D^{0\text{-}1,c}[\text{Thresh}_{D,f,c} \circ f] + \text{er}_D^{0\text{-}1,c}[\text{Thresh}_{D,f,c} \circ f] - \text{er}_D^{0\text{-}1,c,*} \\
&\hspace{4cm} (\text{where } \text{Thresh}_{D,f,c} \text{ is obtained from (OP1)}) \\
&= \left( \text{er}_D^{0\text{-}1,c}[\text{sign} \circ (f - \widehat{t}_{S,f,c})] - \text{er}_D^{0\text{-}1,c}[\text{Thresh}_{D,f,c} \circ f] \right) + \text{regret}_D^{0\text{-}1,c}[\text{Thresh}_{D,f,c} \circ f].
\end{aligned}$$
$$(4)$$

The second term in the above expression can be upper bounded in terms of the ranking regret of $f$ using Theorem 4. We now derive a bound on the first term by using standard VC-dimension based uniform convergence result for binary classification. Note that the real-valued function $f$, when applied to each instance drawn from $D$, induces a distribution over $\mathbb{R} \times \{\pm 1\}$; let us call this distribution $D_f$. Also, let $S_f = \{(f(x_1), y_1), \ldots, (f(x_n), y_n)\}$ be the set constructed by applying $f$ to each instance in $S$; given that $S$ is drawn iid from $D$, it follows that $S_f$ is also iid drawn from $D_f$. Recall that $\mathcal{T}_{\text{inc}}$ is the set of all increasing functions from $\mathbb{R}$ to $\{\pm 1\}$ (see Section 3). One can now view the optimization problem in (OP1) as risk minimization over $\mathcal{T}_{\text{inc}}$ w.r.t. the distribution $D_f$ and the optimization problem in (OP2) as empirical risk minimization over $\mathcal{T}_{\text{inc}}$ w.r.t. the training sample $S_f$. In other words,

$$\inf_{\theta \in \mathcal{T}_{\text{inc}}} \left\{ \text{er}_D^{0\text{-}1,c}[\theta \circ f] \right\} = \inf_{\theta \in \mathcal{T}_{\text{inc}}} \left\{ \text{er}_{D_f}^{0\text{-}1,c}[\theta] \right\} = \text{er}_{D_f}^{0\text{-}1,c}[\theta^*]$$

and

$$\inf_{t \in \mathbb{R}} \left\{ \text{er}_S^{0\text{-}1,c}[\text{sign} \circ (f - t)] \right\} = \inf_{\theta \in \mathcal{T}_{\text{inc}}} \left\{ \text{er}_{S_f}^{0\text{-}1,c}[\theta] \right\} = \text{er}_{S_f}^{0\text{-}1,c}[\widehat{\theta}].$$

Thus the first term in Eq. (4) evaluates to $\text{er}_{D_f}^{0\text{-}1,c}[\widehat{\theta}] - \text{er}_{D_f}^{0\text{-}1,c}[\theta^*]$. Using standard results, one can show that the following upper bound on this quantity holds with probability at least $1 - \delta$ (over the draw of $S \sim D^n$):

$$\text{er}_{D_f}^{0\text{-}1,c}[\widehat{\theta}] - \text{er}_{D_f}^{0\text{-}1,c}[\theta^*] \leq \sqrt{\frac{32\big(\text{VC-dim}(\mathcal{T}_{\text{inc}})\big(\ln(2n) + 1\big) + \ln\big(\frac{4}{\delta}\big)\big)}{n}},$$

where VC-dim($\mathcal{T}_{\text{inc}}$) is the VC dimension of $\mathcal{T}_{\text{inc}}$. Thus with probability at least $1 - \delta$ (over the draw of $S \sim D^n$), we have

$$\begin{aligned}
&\text{regret}_D^{0\text{-}1,c}[\text{sign} \circ (f - \widehat{t}_{S,f,c})] \\
&\leq \sqrt{\frac{32\big(\text{VC-dim}(\mathcal{T}_{\text{inc}})\big(\ln(2n) + 1\big) + \ln\big(\frac{4}{\delta}\big)\big)}{n}} + \sqrt{2}\sqrt{p(1-p)\,\text{regret}_D^{\text{rank}}[f]}.
\end{aligned}$$

It is easy to see that VC-dim($\mathcal{T}_{\text{inc}}$) = 2; plugging this in the above expression completes the proof. $\qquad\square$

# C  Proof of Theorem 10

Our proof for Theorem 10 is simpler than the one in [20] which holds for a more general result. We first state and prove two lemmas which will be useful in our proof.

**Lemma 20.** *Let $D$ be a distribution over $X \times \{\pm 1\}$. For any binary class probability estimator $\widehat{\eta} : X \to [0,1]$ calibrated w.r.t. $D$ and threshold $t \in [0,1]$,*

$$\text{er}_D^{0\text{-}1,c}\big[\text{sign} \circ (\widehat{\eta} - t)\big] = \mathbf{E}_{s_{\widehat{\eta}}}\big[(1-c)s_{\widehat{\eta}}\mathbf{1}(s_{\widehat{\eta}} \leq t) + c\big(1 - s_{\widehat{\eta}}\big)\mathbf{1}(s_{\widehat{\eta}} > t)\big]$$

*and*

$$\text{er}_D^{0\text{-}1,c}\big[\overline{\text{sign}} \circ (\widehat{\eta} - t)\big] = \mathbf{E}_{s_{\widehat{\eta}}}\big[(1-c)s_{\widehat{\eta}}\mathbf{1}(s_{\widehat{\eta}} < t) + c\big(1 - s_{\widehat{\eta}}\big)\mathbf{1}(s_{\widehat{\eta}} \geq t)\big],$$

*where $s_{\widehat{\eta}}$ is the random variable associated with the score distribution of $\widehat{\eta}$ over $[0,1]$.*

*Proof.* We give a proof for the first part of the result; the second part involving $\overline{\text{sign}}$ can be proved in a similar manner. For simplicity of notation, we omit the subscript on $s_{\widehat{\eta}}$. For any $c \in (0,1)$, we have

$$
\begin{aligned}
& \text{er}_D^{\text{0-1},c}\big[\text{sign} \circ (\widehat{\eta} - t)\big] \\
= \ & \mathbf{E}_x\big[(1-c)\eta(x)\mathbf{1}(\widehat{\eta}(x) \leq t) \ + \ c\big(1 - \eta(x)\big)\mathbf{1}(\widehat{\eta}(x) > t)\big] \\
= \ & \mathbf{E}_s\Big[\mathbf{E}_x\big[(1-c)\eta(x)\mathbf{1}(\widehat{\eta}(x) \leq t) \ + \ c\big(1 - \eta(x)\big)\mathbf{1}(\widehat{\eta}(x) > t) \mid \widehat{\eta}(x) = s\big]\Big] \\
= \ & \mathbf{E}_s\Big[(1-c)\mathbf{E}_x\big[\eta(x) \mid \widehat{\eta}(x) = s\big]\mathbf{1}(s \leq t) \ + \ c\big(1 - \mathbf{E}_x\big[\eta(x) \mid \widehat{\eta}(x) = s\big]\big)\mathbf{1}(s > t)\Big] \\
= \ & \mathbf{E}_s\big[(1-c)\mathbf{P}(y = 1|s)\mathbf{1}(s \leq t) + c\big(1 - \mathbf{P}(y = 1|s)\big)\mathbf{1}(s > t)\big] \\
& \hspace{5cm} \text{(follows from } \mathbf{E}_x\big[\eta(x) \mid \widehat{\eta}(x) = s\big] = \mathbf{P}(y = 1|s)). \qquad \square
\end{aligned}
$$

The next lemma states that for any binary class probability estimator $\widehat{\eta}$ calibrated w.r.t. $D$ and a given cost parameter $c \in (0,1)$, the optimal classification transform on $\widehat{\eta}$ that yields minimum cost-sensitive classification error is simply $\theta(u) = \text{sign}(u - c)$.

**Lemma 21.** *Let $D$ be a distribution over $X \times \{\pm 1\}$. For any binary class probability estimator $\widehat{\eta} : X \to [0,1]$ calibrated w.r.t. $D$ and cost parameter $c \in (0,1)$,*
$$
\text{Thresh}_{D,\widehat{\eta},c} = \text{sign} \circ (\widehat{\eta} - c).
$$

*Proof.* Let $s_{\widehat{\eta}}$ denote the random variable associated with the score distribution of $\widehat{\eta}$ over $[0,1]$; for simplicity of notation, we omit the subscript on $s_{\widehat{\eta}}$. Let us start by considering functions $\theta \in \mathcal{T}_{\text{inc}}$ of the form $\theta(u) = \text{sign}(u - t)$ for some $t \in [0,1]$. For any $c \in (0,1)$, we have

$$
\begin{aligned}
& \text{argmin}_{t \in [0,1]}\Big\{\text{er}_D^{\text{0-1},c}\big[\text{sign} \circ (\widehat{\eta} - t)\big]\Big\} \\
= \ & \text{argmin}_{t \in [0,1]}\Big\{\mathbf{E}_s\big[\underbrace{(1-c)s\mathbf{1}(s \leq t) + c\big(1 - s\big)\mathbf{1}(s > t)}_{\text{minimum at } t = c}\big]\Big\} \quad \text{(from Lemma 20)} \\
= \ & c.
\end{aligned}
$$

The last step follows from the fact that the point-wise minimum is attained at $t = c$; this implies that $\theta(u) = \text{sign}(u - c)$ yields the least possible value of $\text{er}_D^{\text{0-1},c}\big[\theta \circ \widehat{\eta}\big]$ over all increasing functions in $\mathcal{T}_{\text{inc}}$, and hence we have $\text{Thresh}_{D,\widehat{\eta},c} = \text{sign} \circ (\widehat{\eta} - c)$. $\qquad \square$

We are now ready to prove Theorem 10. As before, let $s_{\widehat{\eta}}$ denote the random variable associated with the score distribution of $\widehat{\eta}$ over $[0,1]$; for simplicity of notation, let us omit the subscript on $s_{\widehat{\eta}}$.

*Proof of Theorem 10.* Starting with the right hand side, we have

$$
\begin{aligned}
& 2\mathbf{E}_{c \sim U(0,1)}\big[\text{er}_D^{\text{0-1},c}\big[\text{Thresh}_{D,f,c} \circ f\big]\big] \\
= \ & 2\mathbf{E}_{c \sim U(0,1)}\big[\text{er}_D^{\text{0-1},c}\big[\text{sign} \circ (\widehat{\eta} - c)\big]\big] \quad \text{(from Lemma 21)} \\
= \ & 2\mathbf{E}_{c \sim U(0,1)}\Big[\mathbf{E}_s\big[(1-c)s\mathbf{1}(s \leq c) + c(1 - s)\mathbf{1}(s > c)\big]\Big] \quad \text{(from Lemma 20)} \\
= \ & 2\mathbf{E}_s\Big[\mathbf{E}_{c \sim U(0,1)}\big[(1-c)s\mathbf{1}(s \leq c)\big] + \mathbf{E}_{c \sim U(0,1)}\big[c(1 - s)\mathbf{1}(s > c)\big]\Big] \\
& \hspace{6cm} \text{(exchanging expectations)} \\
= \ & 2\mathbf{E}_s\Big[s\int_s^1 (1-c)\,dc + (1-s)\int_0^s c\,dc\Big] \\
= \ & \mathbf{E}_s\big[s(1-s)^2 + (1-s)s^2\big] \\
= \ & \mathbf{E}_s\big[\mathbf{P}(y = 1|s)(1-s)^2 + \big(1 - \mathbf{P}(y = 1|s)\big)s^2\big] \quad \text{(since } \widehat{\eta} \text{ is calibrated)} \\
= \ & \mathbf{E}_x\big[\eta(x)(1 - \widehat{\eta}(x))^2 + \big(1 - \eta(x)\big)\widehat{\eta}(x)^2\big] \\
& \hspace{4cm} \text{(follows from } \mathbf{P}(y = 1|s) = \mathbf{E}_x\big[\eta(x) \mid \widehat{\eta}(x) = s\big]) \\
= \ & \text{er}_D^{\text{sq}}[\widehat{\eta}].
\end{aligned}
$$

$\qquad \square$

# D  Proof of Lemma 11

*Proof.* Expanding the left hand side, we have

$$
\begin{aligned}
\text{regret}_D^{\text{sq}}[\widehat{\eta}] &= \text{er}_D^{\text{sq}}[\widehat{\eta}] - \text{er}_D^{\text{sq},*} = \text{er}_D^{\text{sq}}[\widehat{\eta}] - \text{er}_D^{\text{sq}}[\eta] \\
&= 2\mathbf{E}_{c\sim U(0,1)}\big[\text{er}_D^{\text{0-1},c}\big[\text{Thresh}_{D,\widehat{\eta},c}\circ\widehat{\eta}\big]\big] - 2\mathbf{E}_{c\sim U(0,1)}\big[\text{er}_D^{\text{0-1},c}\big[\text{Thresh}_{D,\eta,c}\circ\eta\big]\big] \\
&\qquad\qquad\qquad\text{(from Theorem 10)} \\
&= 2\mathbf{E}_{c\sim U(0,1)}\big[\text{er}_D^{\text{0-1},c}\big[\text{Thresh}_{D,\widehat{\eta},c}\circ\widehat{\eta}\big]\big] - 2\mathbf{E}_{c\sim U(0,1)}\big[\text{er}_D^{\text{0-1},c}\big[\text{sign}\circ(\eta - c)\big]\big] \\
&\qquad\qquad\qquad\text{(from Lemma 21)} \\
&= 2\mathbf{E}_{c\sim U(0,1)}\big[\text{er}_D^{\text{0-1},c}\big[\text{Thresh}_{D,\widehat{\eta},c}\circ\widehat{\eta}\big] - \text{er}_D^{\text{0-1},c,*}\big] \\
&\leq \sqrt{8p(1-p)\,\text{regret}_D^{\text{rank}}[\widehat{\eta}]} \quad \text{(from Theorem 4).}
\end{aligned}
$$

$\square$

# E  Proof of Lemma 13

We will find it useful to introduce a few notations. For a given ranking model $f : X \to [a, b]$ and distribution $D$ over $X \times \{\pm 1\}$, define $\bar{\mu}_f(t) = \mathbf{P}(f(x) \leq t)$ and $\bar{\eta}_f(t) = \mathbf{P}(y = 1, f(x) \leq t)$ for all $t \in [a, b]$; as before, $p = \mathbf{P}(y = 1)$.

We first state a result of [27, 28] that characterizes the minimizer of (OP3).

**Theorem 22** ( [27, 28]). *Let $f : X \to [a, b]$ (where $a, b \in \mathbb{R}$, $a < b$) be any bounded-range ranking model and $D$ be any probability distribution over $X \times \{\pm 1\}$ such that $(D, f)$ satisfies Assumption A. Moreover assume that $\mu_f$ (see Assumption A), if mixed, does not have a point mass at the end-points $a, b$, and that the function $\eta_f : [a, b] \to [0, 1]$ defined as $\eta_f(t) = \mathbf{P}(y = 1 \,|\, f(x) = t)$ is square-integrable w.r.t. the density of the continuous part of $\mu_f$. Then the minimizer $\text{Cal}_{D,f} : [a, b] \to [0, 1]$ of (OP3) exists, and $\text{Cal}_{D,f}(\tau)$ for any $\tau \in (a, b)$ is given by the right-continuous slope of the largest convex minorant[5] of following graph at $t = \tau$:*

$$
G[f] = \big\{\big(\bar{\mu}_f(t),\ \bar{\eta}_f(t)\big) \ :\ t \in [a, b]\big\}. \tag{5}
$$

*Moreover, $G[\text{Cal}_{D,f} \circ f]$ is piece-wise linear on all portions where it disagrees with $G[f]$; in particular, there exists a collection of disjoint open intervals $\{(a_\alpha, b_\alpha) \,|\, \alpha \in \Lambda\}$ in $[a, b]$, where $\Lambda$ is some index set, such that $\text{Cal}_{D,f}$ evaluates to a constant on each such interval (with the constant being distinct for each interval) and $\text{Cal}_{D,f}$ is equal to $\eta_f$ everywhere else in $[a, b]$:*

$$
\text{Cal}_{D,f}(t) = \begin{cases} \nu_\alpha & \text{if } t \in (a_\alpha, b_\alpha), \text{ for some } \alpha \in \Lambda \\ \eta_f(t) & \text{otherwise} \end{cases},
$$

*where*

$$
\nu_\alpha = \frac{\bar{\eta}_f(b_\alpha) - \bar{\eta}_f(a_\alpha)}{\bar{\mu}_f(b_\alpha) - \bar{\mu}_f(a_\alpha)}, \tag{6}
$$

*with $\nu_\alpha \neq \nu_{\alpha'}$ for any $\alpha \neq \alpha'$, $\alpha, \alpha' \in \Lambda$.*

While the proof for the above result in [27, 28] assumes a continuous and strictly positive density $\mu_f$ over $[a, b]$, it can be extended to handle the slightly more general conditions considered here.

We are now ready to prove the two properties stated for $\text{Cal}_{D,f}$ in Lemma 13.

*Proof of Lemma 13.* We shall assume that the score distribution of $f$ over $[a, b]$ is continuous, and $\mu_f$ denotes the corresponding probability density function; a similar proof can be shown when the score distribution is discrete or is mixed and satisfies conditions stated in the Lemma. For simplicity of notation, let us denote $\text{Cal}_{D,f}$ as Cal.

*Proof of (1):* We need to show that for any $u \in \text{range}(\text{Cal} \circ f)$, $\mathbf{P}(y = 1 \,|\, \text{Cal}(f(x)) = u) = u$. There are three possible cases that we could consider: (i) $u = \nu_\alpha$, for some unique $\alpha \in \Lambda$ (see

Eq. (6)), with $\mathrm{Cal}(t) = u, \forall t \in (a_\alpha, b_\alpha)$, and $\mathrm{Cal}(t) \neq u$, for all $t \notin (a_\alpha, b_\alpha)$; (ii) $u \neq \nu_\alpha$, for any $\alpha \in \Lambda$; (iii) $u = \nu_\alpha$ for some unique $\alpha \in \Lambda$, and there exists $t \notin \cup_{\alpha \in \Lambda}(a_\alpha, b_\alpha)$ with $\mathrm{Cal}(t) = u$.

For any $u \in \mathrm{range}(\mathrm{Cal} \circ f)$ satisfying case (i), there exists $\alpha \in \Lambda$ s.t. $\nu_\alpha = u$. We have from Eq. (6),

$$
\begin{aligned}
u &= \frac{\bar{\eta}_f(b_\alpha) - \bar{\eta}_f(a_\alpha)}{\bar{\mu}_f(b_\alpha) - \bar{\mu}_f(a_\alpha)} \\
&= \frac{\int_{a_\alpha}^{b_\alpha} \eta_f(s)\mu_f(s)ds}{\int_{a_\alpha}^{b_\alpha} \mu_f(s)ds} \\
&= \mathbf{P}\big(y = 1 \mid f(x) \in (a_\alpha, b_\alpha)\big) \\
&= \mathbf{P}\big(y = 1 \mid \mathrm{Cal}(f(x)) = u\big).
\end{aligned}
$$

The last step follows from the fact that for all $t \notin (a_\alpha, b_\alpha)$, $\mathrm{Cal}(t) \neq u$.

For any $u \in \mathrm{range}(\mathrm{Cal} \circ f)$ satisfying case (ii), there exists no $\alpha \in \Lambda$ with $\nu_\alpha = u$; we thus have from Theorem 22 that $\eta_f(t) = u$ for all $t$ with $\mathrm{Cal}(t) = u$. Then

$$
\begin{aligned}
\mathbf{P}\big(y = 1 \mid \mathrm{Cal}(f(x)) = u\big) &= \frac{\int_{\{s \,:\, \mathrm{Cal}(s)=u\}} \eta_f(s)\mu_f(s)ds}{\int_{\{s \,:\, \mathrm{Cal}(s)=u\}} \mu_f(s)ds} \\
&= \frac{\int_{\{s \,:\, \mathrm{Cal}(s)=u\}} u\mu_f(s)ds}{\int_{\{s \,:\, \mathrm{Cal}(s)=u\}} \mu_f(s)ds} \\
&= u.
\end{aligned}
$$

For any $u \in \mathrm{range}(\mathrm{Cal} \circ f)$ satisfying case (iii), there exists a unique $\alpha \in \Lambda$ for which $\nu_\alpha = u$, with $\mathrm{Cal}(t) = u, \forall t \in (a_\alpha, b_\alpha)$, and there also exists $t \notin \cup_{\alpha \in \Lambda}(a_\alpha, b_\alpha)$, for which $\mathrm{Cal}(t) = \eta_f(t) = u$.

$$
\begin{aligned}
\mathbf{P}\big(y = 1 \mid \mathrm{Cal}(f(x)) = u\big) &= \frac{\int_{\{s \,:\, \mathrm{Cal}(s)=u\}} \eta_f(s)\mu_f(s)ds}{\int_{\{s \,:\, \mathrm{Cal}(s)=u\}} \mu_f(s)ds} \\
&= \frac{\int_{a_\alpha}^{b_\alpha} \eta_f(s)\mu_f(s)ds \; + \; \int_{\{s \,:\, \mathrm{Cal}(s)=\eta_f(s)=u\}} \eta_f(s)\mu_f(s)ds}{\int_{\{s \,:\, \mathrm{Cal}(s)=u\}} \mu_f(s)ds} \\
&= \frac{u\int_{a_\alpha}^{b_\alpha} \mu_f(s)ds \; + \; u\int_{\{s \,:\, \mathrm{Cal}(s)=\eta_f(s)=u\}} \mu_f(s)ds}{\int_{\{s \,:\, \mathrm{Cal}(s)=u\}} \mu_f(s)ds} \\
&\qquad\text{(applying Eq. (6) to the first integral in the numerator)} \\
&= u.
\end{aligned}
$$

*Proof of (2):* Recall that for a ranking model $f$, $\mathrm{er}_D^{\mathrm{rank}}[f]$ is equivalent to one minus the area under the ROC curve[6] (AUC) of $f$. It is thus enough to show that the ROC curve of $\mathrm{Cal} \circ f$ is a majorant for the ROC curve of $f$. The ROC curve for $f$ can be defined as

$$
\begin{aligned}
\mathrm{ROC}[f] &= \left\{ \Big( \mathbf{P}(f(x) \leq t \mid y = -1), \, \mathbf{P}(f(x) > t \mid y = 1) \Big) \,:\, t \in [a, b] \right\} \\
&= \left\{ \left( \frac{1}{1-p}\int_a^t (1 - \eta_f(s))\mu_f(s)ds, \, \frac{1}{p}\int_t^b \eta_f(s)\mu_f(s)ds \right) \,:\, t \in [a, b] \right\}. \quad (7)
\end{aligned}
$$

As illustrated in Figure 4, each point in the graph $G[f]$ (defined in Eq. (5)) has a corresponding point in $\mathrm{ROC}[f]$; similarly, each line segment in $G[f]$ corresponds to a line segment in $\mathrm{ROC}[f]$. Moreover, for any two given ranking models $f_1$ and $f_2$, if a line segment in $G[f_1]$ is a minorant for a certain portion of $G[f_2]$, the corresponding line segment in $\mathrm{ROC}[f_1]$ is a majorant for the corresponding portion of $\mathrm{ROC}[f_2]$ (see segments AB and A'B' in Figure 4). Since, from Theorem 22, we have that $G[\mathrm{Cal} \circ f]$ is a minorant for $G[f]$, and $G[\mathrm{Cal} \circ f]$ is piece-wise linear on all portions where it disagrees with $G[f]$, it follows that $\mathrm{ROC}[\mathrm{Cal} \circ f]$ is a majorant for $\mathrm{ROC}[f]$. $\qquad\square$

(a) $G[f]$               (b) ROC$[f]$

Figure 4: Sample plots illustrating the relationship between the graph $G$ (plot of $\bar{\eta}_f(t)$ against $\bar{\mu}_f(t)$ for all $t \in [a,b]$; see Eq. (5)) and the ROC curve (plot of true positive rate $\text{TPR}_f(t) = \mathbf{P}(f(x) > t \mid y = 1)$ against false positive rate $\text{FPR}_f(t) = \mathbf{P}(f(x) \le t \mid y = -1)$ for all $t \in [a,b]$; see Eq. (7)). (a) Graph $G$ for ranking models $f_1$ and $f_2$: the graphs for $f_1$ and $f_2$ agree on all points except for the portion between points $A$ and $B$, where the line segment AB in $G[f_2]$ is a minorant for $G[f_1]$. (b) ROC curve for the ranking models $f_1$ and $f_2$: the points A, B and C in the graph $G$ for $f_1$ and $f_2$ correspond to points A', B' and C' respectively in the ROC curves for $f_1$ and $f_2$; the line segment AB in $G[f_2]$ corresponds to the line segment A'B' in ROC$[f_2]$, which is a majorant for the corresponding portion in ROC$[f_1]$. Moreover, while $G[f_2]$ is a convex minorant for $G[f_1]$, the corresponding ROC curve ROC$[f_2]$ is a concave majorant for ROC$[f_1]$.

## F    Proof of Theorem 14

*Proof.* Using the fact that $\text{Cal}_{D,f} \circ f$ is calibrated (property 1 in Lemma 13), we have

$$
\begin{aligned}
\text{regret}_D^{\text{sq}}[\text{Cal} \circ f] &\le \sqrt{8p(1-p)\,\text{regret}_D^{\text{rank}}[\text{Cal}_{D,f} \circ f]} \quad \text{(from Lemma 11)} \\
&\le \sqrt{8p(1-p)\,\text{regret}_D^{\text{rank}}[f]} \quad \text{(from property 2 in Lemma 13)}.
\end{aligned}
$$

$\square$

## G    Proof of Theorem 16

*Proof.*

$$
\begin{aligned}
\text{regret}_D^{\text{sq}}[\widehat{\text{Cal}}_{S,f} \circ f] &= \text{er}_D^{\text{sq}}[\widehat{\text{Cal}}_{S,f} \circ f] - \text{er}_D^{\text{sq}}[\eta] \\
&= \text{er}_D^{\text{sq}}[\widehat{\text{Cal}}_{S,f} \circ f] - \text{er}_D^{\text{sq}}[\text{Cal}_{D,f} \circ f] + \text{er}_D^{\text{sq}}[\text{Cal}_{D,f} \circ f] - \text{er}_D^{\text{sq}}[\eta] \\
&= \left( \text{er}_D^{\text{sq}}[\widehat{\text{Cal}}_{S,f} \circ f] - \text{er}_D^{\text{sq}}[\text{Cal}_{D,f} \circ f] \right) + \text{regret}_D^{\text{sq}}[\text{Cal}_{D,f} \circ f] \quad (8)
\end{aligned}
$$

Using Theorem 14, the second term in the above expression can be upper bounded in terms of the ranking regret of $f$. We now focus on upper bounding the first term. As in the proof of Theorem 6, consider the distribution $D_f$ induced by $f$ over $\mathbb{R} \times \{\pm 1\}$ and let $S_f$ be the set obtained by applying $f$ to each instance in $S$; clearly, $S_f$ is iid drawn from $D_f$. One can then view the optimization problem in OP4 as empirical risk minimization over $\mathcal{G}_{\text{inc}}$ w.r.t. the sample $S_f$. Using standard Rademacher averages based uniform convergence result for empirical risk minimization over a real-valued function class with the squared loss, we have that the following holds with probability at least $1 - \delta$ (over the draw of $S \sim D^n$):

$$
\text{er}_D^{\text{sq}}[\widehat{\text{Cal}}_{S,f} \circ f] - \inf_{g \in \mathcal{G}_{\text{inc}}} \text{er}_D^{\text{sq}}[g \circ f] \le 4R_{S_f}(\mathcal{G}_{\text{inc}}) + 2\sqrt{\frac{2\ln\left(\frac{8}{\delta}\right)}{n}},
$$

where $R_{S_f}(\mathcal{G}_{\text{inc}})$ is the empirical Rademacher average of $\mathcal{G}_{\text{inc}}$ w.r.t. $S_f$. Using Dudley's integral, and bounds on covering numbers of $\mathcal{G}_{\text{inc}}$, one can show $R_{S_f}(\mathcal{G}_{\text{inc}}) \le 24\sqrt{\frac{2\ln(n)}{n}}$ (see for example [21]);

we thus have with probability at least $1 - \delta$ (over the draw of $S \sim D^n$),

$$\mathrm{er}^{\mathrm{sq}}_D[\widehat{\mathrm{Cal}}_{S,f} \circ f] - \inf_{g \in \mathscr{G}_{\mathrm{inc}}} \mathrm{er}^{\mathrm{sq}}_D[g \circ f] \leq 96\sqrt{\frac{2\ln(n)}{n}} + 2\sqrt{\frac{2\ln\left(\frac{8}{\delta}\right)}{n}}.$$

Plugging this into Eq. (8) (along with the upper bound on the second term) completes the proof. $\square$

## Footnotes

[5]A real-valued function $g_1$ is a minorant of another real-valued function $g_2$ defined over the same domain, if $g_1(z) \leq g_2(z)$, $\forall z$; similarly, $g_1$ is a majorant of $g_2$, if $g_1(z) \geq g_2(z)$, $\forall z$.

[6]The ROC curve of a ranking model $f$ is the plot of the true positive rate (probability of classifying a random positive example as positive) against the false positive rate (probability of classifying a random negative example as positive) of a classifier of the form $\mathrm{sign} \circ (f - t)$ for all thresholds $t \in [a, b]$.