[Reviews · NeurIPS 2013]

Submitted by Assigned_Reviewer_4

The results in this paper establish new relationships between binary classification (C), class probability estimation (P), and bipartite ranking (R). Previously existing results had shown that it is possible to solve C problems by thresholding estimators for P problems and that R problems can be solved via P problems. By establishing regret bounds between problems (i.e., showing that if the regret on problem A is small then small regret on problem B is too) this paper adds to these relationships by showing that ranking algorithms can be used to solve both C and P problems. To do so required the introduction of "weak regret transfer bounds" in which a critical parameter in the reductions (e.g., a threshold) depends on the unknown data distribution. They also show how empirical estimates of these parameters can be used effectively.

The work in this paper is of a high standard. All three of these problems are increasingly well studied within the machine learning community so any clarification of their relationships is of significance. The presentation is very clear, well motivated, and easy to read.

One slight complaint is that the paper does not discuss existing relationships between *families* of classification problems and probability estimation, such as the probing reduction of Langford and Zadrozny ("Estimating class membership probabilities using classifier learners", AISTATS 2005). Although this is not directly related to the type of question addressed by the paper it seems relevant.
Summary: The results in the paper address a clear gap in existing results connecting classification, probability estimation, and ranking by introducing a novel notion of "weak regret transfer bounds". These results are significant and very well presented.

Submitted by Assigned_Reviewer_5

Summary.

In more detail, this paper considers BC (binary classification), BBR (binary bipartite ranking -- elements of one class should all be ranked ahead of the other class), and CPE (binary class probability estimation). As discussed in the short review, it is shown that a consistent method for BBR, taken as a black box, can be used to produce a consistent method for CPE and BC. The key technical point is that the rankings must be "calibrated": a sample must be used to either figure out where the positive/negative threshold should lie to solve BC, or, in the more refined case of CPE, isotonic regression is used to find the univariate monotonic map from ranking scores to probability estimates.

This paper provides these reductions (first with idealized distributional access, and then via finite sample estimates), and closes with some experiments which show that if one has an existing BBR model and some extra samples, then it is better to train the specified reductions, rather than just using the sample to train a CPE or BC model from scratch. While the experimentation is not extensive, this is exactly the sort of sanity check one would like to see.

I will note that the use of extra samples to calibrate a reduction of this type appears to be new (that is, existing reductions which, say, solve BC and BBR via CPE, do not need to use extra samples).


Evaulation.

While perhaps not the world's most glamorous problem, it is foundational, and I am happy to see progress. The solutions given above are sketched in the body, and it is intuitive that some extra data is needed to adapt the ranking values.

I like the paper, but see one weakness. The definition of reduction ("weak regret transfer bound") makes no restrictions on how the new samples are used; in particular, whenever class B of problems is learnable, then it is possible to reduce from B to any class A simply by ignoring A and using the fresh samples to learn B directly.

Personally, I would like the authors to add some writing to address the following points. To be clear, no request is being made for new theorems or extensive extra work; simply I think the writing can both clarify the above issue, and also pave the way for future work to deal with it rigorously.

1. It should be stressed that reductions should not be trivial in the sense above (ignoring the provided ranking), and moreover that the two presented in this paper are not trivial in that way. Moreover, the theorem statements in this paper directly provide the reduction, and thus the reader can verify there is no cheating. I note that the simulation section is a comparison to the start-from-scratch approach, and thus the authors are aware of this issue, but I think it deserves significant mention well before that section.

2. It might be nice to explicitly state as an open problem the task of formalizing the notion of "no cheating", for instance some sort of information theoretic approach that asserts these reductions can use fewer samples than what you'd need if provided only fresh examples and no ranking function.

3. As a step in this direction, it may be worth stressing that the sample complexity of the binary classification reduction is very nice, only paying for thresholds, and not for the full price of learning within some potentially large class. Unfortunately, on the other hand, the probability estimation reduction pays the full price of isotron, which appears unnecessary since only the link function and not the direction are used. Thus, another open problem may be reducing the sample complexity in the reduction from probability estimation.

One more comment, not along the lines of the above "weakness". It would be nice if there were at least a heuristic argument that extra examples are necessary (i.e., "strong reductions" are impossible). For classification, I think it suffices to fix a ranking and show that, without extra samples, one can not decide which of two classifications (via thresholding) are better, and thus samples are necessary. For probability estimation, similarly considering two different links (from a fixed ranking) could work.


Detailed/Minor feedback.

Introduction. I think the intro can more cleanly identify the core strategy of the paper. For example, by my reading, the basic outline is (1) these binary problems all seem related, but we currently have gaps, then (2) the problem is that BBR scores can be stretched out without tweaking the loss, which is not true for BC and CPE, (3) so we should use a finite sample / distributional access to adapt them, (4) for BC this is just a threshold and for CPE it's just the link part of isotonic regression. If the authors like this approach, I think honoring it could also hint how to break the long paragraph in lines 64-81.
Line 25. I think the abstract could be made more accessible either by defining "Regret transfer bound" or by replacing it with an explanation.
Line 29. I am not sure this is surprising.
Line 43. I suggest also citing Tong Zhang's classification consistency paper, which Bartlett/Jordan/McAuliffe cite heavily, and which appears in for instance the Boucheron-Bousquet-Lugosi survey as the origination of these rigorous convex -> zero/one bounds.
Line 72. I think this would be a good place to discuss the necessity of fresh samples (as discussed above).
Line 79. Regarding citations for this covering bound (which follows by gridding the domain and range?), there are is probably earlier work (by a number of decades) in the analysis literature...?
Line 89. This is nice and I think it could be stressed.
Line 125. I guess f should map to \bar R, but isotonic regression doesn't handle this as far as I know, so I recommend switching to R.
Line 177. I am not sure the minimizer exists for arbitrary f. In particular, imagine that error decreases as t approaches some t0, but then jumps up at t0. This should be possible when the pushforward measure through f has a point mass at t0. This sort of thing is abusing the fact that "sign" function decides 0 is positive.
Line 215. Rather than "equal parts", might be good to stress you can get away with using far fewer samples for the reduction step.
Line 219. I think this is a nice point.
Line 269. Similarly to Line 177, I wonder if this minimizer exists. Also, though I didn't check your details carefully, I suppose your PAV procedure may be working with Lipschitz functions (or at least continuous), in which case this really may not be attained. I realized you have a reference here for the minimum being attained, but that is a 57 page paper and I must admit I did not check for distributional assumptions (ruling out the sorts of non-Lebesgue-continuous measures I mention in Line 177).
Line 278. I suggest parentheses or something around OP3 so it's clearly referring to line 269.
Line 297. It might be good to mention isotron earlier, for instance in the intro as I dicsussed above.
Line 307. As mentioned throughout, I think some discussion of the n^{-1/3} is warranted. If you believe the bound is not tight, maybe state improvements as an open problem.
Line 341. Just like Remark 8, remark 19 is very nice (note, I think the last sentence went astray?).
Line 504. Missing an [h] to the left of the leftmost equality.
Line 602. I guess this 3 is "definition 3"; similar issues on Line 615.5.

Summary: This paper shows how ranking problems can solve both probability estimation problems and binary classification problems; in particular, efficient algorithmic reductions are given whereby a consistent ranking algorithm leads to consistent probability estimation and consistent classification.

I think this is a foundational problem, and am glad to see it worked out; moreover, the paper is clearly written, with the mechanisms and key elements of the reductions explained throughout the body (all proofs appear in appendices). My only misgiving is that the reductions require a fresh sample, and it is not clear how much this is necessary.

Submitted by Assigned_Reviewer_6

The paper shows that a bound on the quality of a bipartite ranking can be transferred to a bound on classification and also to a bound on probability estimation. These results are not all that surprising nor are the proofs hard to obtain, but the paper is very well written. I thus think that the papers makes a good contribution.

Much of the paper seems to build upon reference [18]. What exactly is new here?

The authors may wish to look at multipartite rankings, which are somewhere inbetween bipartite rankings and probability estimates.
Summary: This is a very well written paper that seems to close a gap in the literature.
Author Feedback

Author rebuttal: Thanks to all the reviewers for their careful reading and positive feedback. Below are responses to the main questions/comments.

Reviewer 4

- Probing reduction of Langford & Zadrozny: Certainly, we will include a pointer to this.

Reviewer 5

- Definition of reduction: Yes, this is a good point. We will clarify that such reductions are useful only when the number of samples needed is smaller than when starting from scratch. As you note, the binary classification reduction indeed satisfies this criterion; we will make this explicit.

- "It would be nice if there were at least a heuristic argument that extra examples are necessary (i.e., "strong reductions" are impossible)" / "My only misgiving is that the reductions require a fresh sample, and it is not clear how much this is necessary":
Please note that in our setting, we start with a given (fixed) ranking function (which may or may not have been learned from some training sample), and aim to convert this to a classification/CPE model. In general, this indeed requires access to the distribution via a sample; please see the example at the end of this response. However, if the ranking function one starts with is learned from a training sample, then it is conceivable that the same sample could be used to construct a suitable classification threshold or CPE model; we leave this as an open problem.

- Detailed/minor feedback: Many thanks for the detailed comments. These will certainly help us improve and tighten the paper.

Reviewer 6

- "Much of the paper seems to build upon reference [18]. What exactly is new here?":
Please note that our focus is on establishing regret transfer bounds from ranking to classification/CPE. Reference [18] shows that various performance metrics used for evaluating CPE models can be viewed as an expected loss (over uniformly drawn misclassification costs) of a classifier built using different threshold choice methods. The results in [18] *do not* amount to a regret transfer bound from ranking to classification/CPE. Indeed, the only main result we borrow from [18] is the one showing equivalence between the expected squared error of a calibrated CPE model and expected loss (over uniformly drawn misclassification costs) of a classifier obtained by choosing the optimal threshold on the model (this follows from Theorem 29 in [18], and is stated as Theorem 10 in our paper). Note that this result is just one step in establishing the regret transfer bound from ranking to CPE.

- "The authors may wish to look at multipartite rankings, which are somewhere inbetween bipartite rankings and probability estimates.":
In our work we consider a setting with binary labels. It would certainly be interesting to look at a multiclass setting, where one can explore similar reductions from multipartite ranking to multiclass classification/CPE.

--

Example showing necessity of access to the distribution (or a sample from it) when starting with a fixed ranking function:

We illustrate with an example that if did not have access to additional examples (and thus to the underlying distribution), a (distribution-free) strong reduction from ranking to classification/CPE is not possible.

Instance space:
X = {x1, x2, x3, x4, x5}

Raking function f:
x1 x2 x3 x4 x5
2 4 5 7 9

Suppose we want to build a binary classifier from the given ranking function. Assume c = 0.5 (equal misclassification costs).

Consider two distribution settings where the given scoring function is a bayes optimal ranking function (a monotonic function of the true class probabilities).

Case 1: (say D1)
Instance distribution:
p(xi) = 1/5
True class probabilities:
\eta(xi) = p(y=1 | xi)
x1 x2 x3 x4 x5
0.1 0.2 0.3 0.4 0.9

Optimal threshold choice (for c=0.5):
h1(x) = sign(f(x) - 8) (threshold value chosen between x4 and x5) is the classifier with min. expected misclassification error over all threshold assignments on f and is is also a bayes optimal classifier w.r.t. D1 with bayes 0-1 error = 0.22.
Clearly, 0-1-regret_D1[h1] = 0.

Case 2: (say D2)
Instance distribution:
p(xi) = 1/5
True class probabilities:
\eta(xi) = p(y=1 | xi)
x1 x2 x3 x4 x5
0.1 0.6 0.7 0.9 0.9

Optimal threshold choice (for c=0.5):
h2(x) = sign(f(x) - 3) (threshold value chosen between x1 and x2) is the classifier with min. expected misclassification error over all threshold assignments on f, as well as a bayes optimal classifier w.r.t. D2 with bayes 0-1 error = 0.2.
Clearly, 0-1-regret_D2[h2] = 0.

Note that in both cases, since f is a bayes optimal ranking function, the ranking regret of f is 0, i.e., ranking-regret_D1[f] = 0 and ranking-regret_D2[f] = 0.

Note that h1 and h2 disagree with each other on 3 examples. It is also easy to see:
0-1-regret_D1[h2] = 0.46 - 0.22 = 0.24 > 0 and
0-1-regret_D2[h1] = 0.48 - 0.20 = 0.28 > 0.

With h1 or any classifier that agrees with h1 on all examples,
0-1-regret_D1[h1] = ranking-regret_D1[f] = 0, while
0-1-regret_D2[h1] > ranking-regret_D2[f] = 0.
With h2 or any classifier that agrees with h2 on all examples,
0-1-regret_D2[h2] = ranking-regret_D2[f] = 0, while
0-1-regret_D1[h2] > ranking-regret_D1[f] = 0.

Clearly, if one did not have access to additional examples (and thus to the underlying distribution) and had to build a classifier from only the given ranking function, a distribution-free strong regret transfer bound guarantee from ranking to classification is not possible.

Also note that in the example given, the same ranking function is bayes optimal for two different distribution settings with different class probabilities. Clearly, if one had to build a CPE model from only the given ranking function, without access to additional examples, a (distribution-free) strong regret transfer bound guaranteed from ranking to CPE is not possible.

--